# Expression of IDO1 and PD-L2 in Patients with Benign Lymphadenopathies and Association with Autoimmune Diseases

**DOI:** 10.3390/biom13020240

**Published:** 2023-01-27

**Authors:** Maysaa Abdulla, Christer Sundström, Cecilia Lindskog, Peter Hollander

**Affiliations:** 1Cancer Immunotherapy, Department of Immunology, Genetics and Pathology, Uppsala University, 75185 Uppsala, Sweden; 2Cancer Precision Medicine, Department of Immunology, Genetics and Pathology, Uppsala University, 75185 Uppsala, Sweden

**Keywords:** IDO1, PD-L2, lymphadenopathy, immunohistochemistry, histopathology

## Abstract

The expression patterns of IDO1 and PD-L2 have not been thoroughly investigated in benign lymphadenopathies. The aim with this study was to elucidate how IDO1 and PD-L2 are expressed in benign lymphadenopathies in patients with autoimmune diseases (AD) compared to patients without AD. Formalin-fixed paraffin-embedded lymph nodes from 22 patients with AD and 57 patients without AD were immunohistochemically stained to detect IDO1 and PD-L2. The material was previously stained with EBER in situ hybridization to detect cells harboring the Epstein–Barr virus (EBV). IDO1 and PD-L2 were generally expressed by leukocytes to low degrees, while follicular IDO1+ cells were very rare. IDO1+ cells in single germinal centers were detected in five patients, and there was a high co-occurrence of follicular EBV+ cells in these cases (three of five patients). There were also significant correlations between interfollicular EBV+ cells and interfollicular IDO1+ cells (Spearman rho = 0.32, *p* = 0.004) and follicular IDO1+ cells (Spearman rho = 0.34, *p* = 0.004). High or low amounts of IDO1+ or PD-L2+ cells were not statistically significantly associated with patients with AD. However, the lymphadenopathy with the highest amount of interfollicular IDO1+ cells, which was also the only lymphadenopathy in which endothelial cells expressed IDO1, was in a patient with sarcoidosis. This study further supports that the EBV induces the expression of IDO1 and our findings should be recognized by future studies on IDO1 and PD-L2 in inflammatory and malignant conditions.

## 1. Introduction

To investigate enlarged lymph nodes in clinical situations of unclear lymphadenopathies, lymph node excision is preferably performed. A lymph node with no evidence of malignancy is generally diagnosed as benign lymphadenopathy [1]. There are numerous reasons for benign lymphadenopathies to develop, e.g., infections and autoimmune diseases [2]. AD is a heterogeneous group of chronic inflammatory conditions, where rheumatoid arthritis (RA) is one of the most prevalent [3]. Patients with RA have an increased risk of developing benign lymphadenopathies, where the enlarged lymph node often is located close to the site of inflammation [4,5]. The reason for the increased risk of developing benign lymphadenopathies in patients with AD is not fully elucidated, but is presumably due to an altered inflammatory activity of the host [2]. 

Indoleamine 2,3 dioxygenase (IDO1) is an enzyme involved in the kynurenine pathway and responsible for degrading tryptophan into L-kynurenine, which suppresses effector T-cells and promotes regulatory T-cells [6], resulting in immune tolerance [7]. Previous studies show that IDO1 is altered in patients with AD [6]. Patients with AD have an increased level of tryptophan catabolism, which indicates high IDO1 activity [8]. High IDO1 is associated with the development of disease symptoms, and IDO1 inhibition resulted in reduced RA symptoms in a mouse model [9]. Therefore, IDO1 has been proposed as a possible target for treatment in AD [10], but expression of IDO1 in AD patients with benign lymphadenopathies has not been investigated. 

Programmed death 1 (PD-1) can be expressed by leukocytes, and is inhibited by other cells that express its ligands (PD-L1 and PD-L2). The PD-1 pathway is important in order to avoid tissue damage induced by T cells that target the host [11]. PD-L1 has been extensively studied in several malignancies [12] and benign conditions such as AD [13], while PD-L2 has been less studied [13]. PD-L1 is a key regulator for peripheral immune tolerance, is induced upon cytokine stimulation (e.g., IFN-y) and expressed by different hematopoietic cells, but also other cells such as endothelium and tumor cells [12], while PD-L2 expression is mainly restricted to antigen-presenting cells (APC), e.g., dendritic cells and macrophages [14]. Thus, PD-L1 has a major role in the maintenance of peripheral tolerance at sites of inflammation, while PD-L2 is mainly restricted to APC and thus is an important immune regulator in lymphoid tissue such as lymph nodes [13]. In systemic lupus erythematosus (SLE), membrane-bound and soluble PD-L2 was associated with disease activity [15]. In another study, PD-L2 was associated with high inflammatory activity and bone loss in RA [16]. PD-L2 expression in benign lymphadenopathies from patients with AD has not been described. 

PD-1 inhibitors are frequently used in different malignancies [17], where treatment resistance and failure to achieve long-lasting remission is common [17,18]. Thus, co-inhibition along with other immune regulatory pathways, e.g., the kynurenine pathway, show promising results [19,20]. However, the increased use of immune checkpoint inhibitors demands a better understanding of the expression of potential targets for treatment, both in the tumor microenvironment of malignancies but also in benign inflammatory conditions [21]. 

As a follow-up to a recent publication from our group [22], where higher proportions of PD-1+ cells and a tendency for lower proportions of PD-L1+ and Epstein–Barr virus (EBV) positive cells in benign lymphadenopathies from patients with AD were found, the aim of this study was to investigate IDO1 and PD-L2 in the same material. This was in order to better understand the immune responses in benign lymphadenopathies, which can be of guidance for further studies on inflammatory and malignant conditions. 

## 2. Materials and Methods

Patients with lymphadenopathies were biopsied due to clinical suspicion of lymphoma, and these were diagnosed as benign lymphadenopathies at the Department of Pathology, Uppsala University Hospital, between 1995 and 2006. A palpable enlarged lymph node was the symptom reported in patients with lymph nodes located in the cervical, axillary, inguinal, breast, face and leg regions. In cases with mediastinal or abdominal lymphadenopathies, cough or abdominal discomfort, respectively, were the symptoms reported. Lymph nodes with available formalin-fixed paraffin-embedded (FFPE) tissue were histopathologically reviewed and included. Each biopsy was classified by one of the authors (PH) based on the predominant lymph node architecture as previously described [22]. Medical records were reviewed and information on diagnosis of AD, treatment of AD, subsequent malignancy and anatomical localization of the lymph node biopsy was collected. The cohort consisted of 22 patients with various AD diagnosed prior to the diagnosis of benign lymphadenopathy. Fifty-seven patients without previous AD prior to the diagnosis of benign lymphadenopathy were selected and matched for age and sex (Figure 1). Patients with chronic infection, organ transplantation, and immunodeficiency syndrome were not included. 

### 2.1. Immunohistochemical Stainings

Four µm FFPE sections were used for immunohistochemical (IHC) stainings. For IDO1, the Autostainer 480 instrument (Thermo Fischer Scientific, Waltham, MA) was used, and for PD-L2, the Intellipath FLX system (Biocare Medical, Pacheco, CA) was used. Rabbit polyclonal antibody HPA023149 (Atlas Antibodies AB, Stockholm, Sweden) (dilution 1:200) was used for IDO1, and rabbit monoclonal antibody D7U8C/82723 (Cell Signaling Technology, Danvers, MA) (dilution 1:50) was used for PD-L2. Antigen retrieval was performed using citrate buffer and TE buffer for IDO1 and PD-L2, respectively. The DAB Qaunto detection kit (Thermo Fischer Scientific, Waltham, MA) and the Betazoid DAB detection kit (Biocare Medical, Walnut Creek, CA) were used to envision IDO1 and PD-L2, respectively. Hematoxylin was utilized to counterstain the slides. Further details regarding IDO1 and PD-L2 stainings have been described previously [23]. EBV-encoded small RNA (EBER) in situ hybridization and IHC stainings for PD-1 and PD-L1 have been previously described in detail [22]. 

### 2.2. Evaluation of IDO1 and PD-L2

IHC stainings for IDO1 and PD-L2 were reviewed manually by an experienced hematopathologist (M.A.). First, the whole lymph node architecture was considered, i.e., all follicular and interfollicular areas were visualized to semi-quantitatively assess the proportions of IDO1+ and PD-L2+ cells. Next, for each case and staining, 5 high power fields (HPF) for follicular areas and 5 HPF for interfollicular areas were chosen. Areas with necrosis and fibrosis were excluded. The HPF representative of the overall amount of positive cells of the lymph node were chosen and quantified at 400x magnification. For IDO1, cells with nuclear expression were considered as positive. For PD-L2, cells with cytoplasmic or membranous staining were considered as positive. Cells with at least weak staining were considered as positive; the staining intensity was not further graded and noted in our evaluation. Cells with no nuclear, membranous or cytoplasmic staining were designated as negative. The number of positive and negative cells was calculated and the proportions of positive cells were calculated in whole percentage increments. In addition, cases with single germinal centers (GC) containing IDO1+ or PDL2+ cells were noted. 

### 2.3. Statistical Methods

To investigate differences in the amounts of IDO1+ and PDL2+ cells between patients with AD and without AD, the Mann–Whitney U-test was used. Tabulated clinicopathological variables were compared with the chi-square or Fischer’s exact test. The cut-offs used for tabulated values were median proportions of follicular and interfollicular IDO1+ and PD-L2+ cells. Correlations between different variables were investigated with the Spearman rank-order correlation test. All tests were two-tailed and *p* values of <0.05 were considered significant. For statistical analyses, R with the Rstudio package 1.3.1093 (www.r-project.org, accessed on 17 September 2020) was used. 

## 3. Results

### 3.1. IDO1 in Follicular and Interfollicular Areas

Based on the morphology of mononuclear cells, the expression of IDO1 was mostly observed in histiocytes with intermediate to strong nuclear expression and often with ramified cytoplasmic projections (Figure 2). Seventeen of seventy-one (24%) patients exhibited some follicular IDO1+ cells while the rest of the patients had no IDO1+ cells. In lymph nodes with IDO1+ cells in follicular areas, the proportion of positive cells were mostly low, while a few cases showed a relatively high amount of IDO1+ cells in several GCs. Five cases contained single GCs with IDO1+ cells, while the rest of the GCs contained no IDO1+ cells (Figure 2D). As for interfollicular areas, the median amount of IDO1+ cells was 5%, and 70 of the 78 evaluable cases (90%) contained interfollicular IDO1+ cells. Most cases contained scattered interfollicular IDO1+ cells, while a few cases showed a high amount of IDO1+ cells in interfollicular regions. A few patients showed abundant IDO1+ cells in interfollicular regions, as well as the expression of IDO1+ cells in several GCs (Figure 3A), while some patients showed a low expression of IDO1 in both follicular and interfollicular regions (Figure 3E). In addition to leukocytes expressing IDO1, we noted one patient with sarcoidosis where endothelial cells expressed IDO1 (Figure 3F). In the same patient, interfollicular EBV+ cells were present and the histologic pattern was paracortical hyperplasia with no signs of granulomatous inflammation.

### 3.2. IDO1 in Single Germinal Centers

Five cases exhibited the expression of IDO1 in single GCs (Figure 2D). This was observed in one patient with AD and four patients without AD. In three of these five patients (60%), follicular EBV+ cells were also observed, while the follicular expression of EBV+ cells were only observed in 7 of 67 (10%) lymph nodes without single GCs containing IDO1+ cells. Of the three cases with single GCs containing IDO1+ cells and the presence of follicular EBV+ cells, two showed follicular hyperplasia and one showed. paracortical hyperplasia. One had a previous AD (autoimmune hepatitis) while two did not have a previous AD, and two were aged <60 and one was aged ≥ 60 years.

### 3.3. PD-L2 in Follicular and Interfollicular Areas

Based on the morphology of mononuclear cells, the expression of PD-L2 was mostly observed in histiocytes with weak to intermediate membranous or cytoplasmic expression and often weakly stained projections (Figure 3). The presence of PD-L2+ cells in follicular areas was mostly low (median 5%), and most GCs contained scattered PD-L2+ cells while some lymph nodes showed a higher abundance of PD-L2+ cells in GCs. Sixty-four of seventy (91%) patients had follicular PD-L2+ cells. Interfollicular PD-L2+ cells were most often scattered and expressed to low degrees (median 5%). Seventy of seventy-eight (90%) patients had interfollicular PD-L2+ cells. Some cases showed a relatively high amount of both follicular and interfollicular PD-L2+ cells (Figure 4A,B), and some cases had a low amount of both follicular and interfollicular PD-L2+ cells (Figure 4C), while some cases only had high interfollicular amounts of PD-L2 (Figure 4D). Interestingly, a few cases showed a partial infiltration of PD-L2+ cells in interfollicular regions (Figure 4A). There were no patients with PD-L2+ cells confined to single GCs. 

### 3.4. Autoimmune Diseases and Association with IDO1 and PD-L2

Twenty-two patients with AD were included, of which ten were diagnosed with RA, three with Sjögren’s syndrome, three SLE, three sarcoidosis, two psoriatic arthritis, one Mb Bechterew, one polymyalgia rheumatica, one primary biliary cirrhosis, one autoimmune hepatitis, and one Mb Still. The median proportions of IDO1+ cells were the same in patients with AD compared to patients without AD in both follicular (0% versus 0%) and interfollicular (5% versus 5%) areas (Figure 4A,B). The median proportions of PD-L2+ cells were slightly higher in patients with AD compared with patients without AD in both follicular (7.5% versus 5%) and interfollicular (5% versus 3%) areas (Figure 4C,D). However, there were no significant differences regarding the follicular and interfollicular expression of IDO1+ cells and PD-L2+ cells between patients with AD and patients without AD using the Mann–Whitney U-test (Figure 4), chi-square, or Fischer’s exact test (Table 1). In additional analyses, patients diagnosed with RA, SLE, or Sjögren´s syndrome (n = 13) were compared with other patients, and patients treated with steroids and/or methotrexate (n = 13) were compared with other patients, but there were no differences in high or low amounts of follicular and interfollicular IDO1+ or PD-L2+ cells in these analyses (Table 1). In Appendix A, only patients with AD were analyzed. There were no significant differences in follicular and interfollicular amounts of IDO1+ cells or PD-L2+ cells when patients with RA; RA and/or SLE; RA and/or Sjögren’s syndrome; or RA, SLE and/or Sjögren’s syndrome were grouped and compared to patients with other AD using Fischer´s exact test (Appendix A). However, we noted that there were only three AD patients with the presence of follicular IDO1+ cells, of which two patients had sarcoidosis and one patient had autoimmune hepatitis.

### 3.5. Histopathologic Pattern and Relation to IDO1 and PD-L2

The most common histopathologic pattern was follicular hyperplasia, followed by paracortical hyperplasia and sinus histiocytosis, while other histopathologic patterns were more uncommon (Table 1). In patients with follicular hyperplasia, there was a significantly higher proportion of patients with high amounts of follicular PD-L2+ cells than low amounts of follicular PD-L2+ cells (55% versus 7%) (Table 1).

### 3.6. Localization of the Lymph Node Biopsy and Relation to IDO1

The most common lymph node biopsy localization was in the cervical region, followed by the axillary and inguinal regions, while other biopsy localizations were less common. In patients with cervical and axillary lymph node biopsies, there were significantly higher proportions of patients with high amounts of interfollicular IDO1+ cells than low amounts of interfollicular IDO1+ cells (41% and 31% versus 26% and 16%, respectively). Inguinal lymph node biopsies were more commonly observed in patients with low amounts of interfollicular IDO1+ cells than high amounts of IDO1+ cells (32% versus 21%) (Table 1).

### 3.7. Subsequent Malignancies and Relation to PD-L2

Seven patients developed subsequent malignancies; diagnoses included three breast carcinomas, two prostate carcinomas, one essential thrombocythemia and one pancreatic carcinoma. The time from lymph node biopsy until diagnosis of malignancy ranged from 10 to 21 years. In patients that developed a subsequent malignancy, there was a slightly higher proportion of patients with high amounts of follicular PD-L2+ cells than low amounts of follicular PD-L2+ cells (19% versus 5%) (Table 1). 

### 3.8. Correlative Analyses 

The Spearman rank order correlation test was used to determine correlations between follicular and interfollicular IDO1+ and PD-L2+ cells. In addition, data on amounts of PD-1+ and PD-L1+ cells and numbers of EBV+ cells were added to the analyses (Table 2). Follicular IDO1+ cells correlated with interfollicular IDO1+ cells (Spearman Rho = 0.39, *p* < 0.001), interfollicular PD-L1+ cells (Spearman Rho = 0.28, *p* = 0.02), and interfollicular EBV+ cells (Spearman Rho = 0.34, *p* = 0.004). Interfollicular IDO1+ cells correlated with interfollicular PD-L1+ cells (Spearman Rho = 0.30, *p* = 0.007) and interfollicular EBV+ cells (Spearman Rho = 0.32, *p* = 0.004). Follicular PD-L2+ cells correlated with follicular PD-L1+ cells (Spearman Rho = 0.37, *p* = 0.002), while interfollicular PD-L2+ cells did not correlate with amounts or numbers of any other cells. In Appendix A, only patients with AD were analyzed (Appendix A), and follicular IDO1+ cells correlated with follicular EBV+ cells (Spearman Rho = 0.61, *p* = 0.007). In patients without AD (Appendix A), no significant correlation was observed between follicular IDO1+ cells and follicular EBV+ cells (Spearman Rho = 0.10, *p* = 0.47).

## 4. Discussion

As a follow-up to our previous study on benign lymphadenopathies in patients with AD compared to patients without AD, we investigated the same material with IHC markers for IDO1 and PD-L2 and related it to other clinicopathological characteristics. When analyzed on a group level, no significant associations between AD and the amounts of IDO1 or PD-L2 were observed. However, when looking further into other results of this study, several intriguing and novel observations were obtained. 

### 4.1. Expression of IDO1 and Relation to EBV 

Follicular expression of IDO1 was rare, while interfollicular IDO1+ cells were more common. There were significant correlations between interfollicular EBV+ cells and both follicular and interfollicular IDO1+ cells. We also observed that IDO1+ cells might be located in single GCs (5 of 79 patients), and there was a strikingly high co-occurrence in these rare cases with a prevalence of follicular EBV+ cells (3 of 5 patients). Our findings imply that EBV might be a contributor to the rare observation of IDO1+ cells confined to single GCs. This might be due to a highly local induction of IDO1 by molecules (e.g., cytokines and chemokines) released due to the presence of EBV [24]. The expression of IDO1 in single GCs was not reported by a previous study that stained benign lymphadenopathies and tumor-draining lymph nodes from patients with melanoma and breast carcinoma [25]. They report that IDO1 was mostly located interfollicularly. No lymphadenopathies contained PD-L2+ cells in single GCs in our study, and IHC stainings for PD-1 and PD-L1 were also reviewed, and we found no patients with single GCs with the expression of PD-1 or PD-L1. The different expression patterns may be due to induction via different cytokines and chemokines from EBV infected cells. Both IDO1 and PD-L1/PD-L2 are induced by IL-6, TNF-a [26], and IFN-y [24,27], while PD-L1 and PD-L2 are also induced by other cytokines, e.g., IL-4 [28]. These observations are intriguing, and to the best of our knowledge, not described previously. Our study further supports the notion that the presence of EBV infected cells leads to the increased expression of IDO1, not only in EBV associated malignancies [24] but also in benign [26] conditions. Lastly, only in patients with AD was there a correlation between follicular IDO1+ cells and follicular EBV+ cells, while this was not observed in patients without AD. Possibly, patients with AD are especially prone to induce IDO1 in GCs when EBV infected cells are present. However, these findings should be interpreted with great caution due to the low number of patients with AD and follicular IDO1+ cells (n = 3). 

### 4.2. IDO1 and PD-L2 and Relation to AD 

IDO1 is expressed mainly by antigen-presenting cells (e.g., dendritic cells and macrophages) in lymph nodes [25]. Previous studies also report that endothelial cells are able to express IDO1 in different malignancies, e.g., melanoma [29] and carcinomas [25] (e.g., cervix and lung). The endothelial expression of IDO1 is mainly induced by IFN-y, which is secreted in the tumor microenvironment [25]. We noted that endothelial cells expressed IDO1; however, this was only observed in one patient with an exceptionally high proportion of interfollicular IDO1+ cells (Figure 3F). This patient had a previous diagnosis of sarcoidosis. Increased levels of CD4+ T-cells producing IFN-y were found in lung lavage [30] and increased levels of IFN-y induced chemokines were observed in serum [31] from patients with sarcoidosis. A previous study reported that patients with lung sarcoidosis have depressed serum tryptophan/kynurenine ratios, indicating an increased activity of IDO1 [32]. Thus, our observation with only one patient with the endothelial expression of IDO1 might be due to the patients’ sarcoidosis. If patients with AD, in addition to an altered PD-1 pathway, have a dysfunctional kynurenine pathway, this should be recognized and further studied and may be of importance in future treatment concepts in AD. Since antirheumatic drugs might alter the immune response in benign lymphadenopathies [33], we investigated if patients treated with steroids and/or methotrexate had higher amounts of IDO1+ or PD-L2+ cells, since these patients might represent a group with high inflammatory activity in their AD. However, we found no association between treatment and amounts of IDO1+ or PD-L2+ cells.

### 4.3. Anatomical Localization of Lymph Node and IDO1 

The observation that patients with high amounts of interfollicular IDO1+ cells more commonly had biopsies from the cervical and axillary regions is also an interesting finding, especially since no certain histological pattern was observed in patients with a high amount of interfollicular IDO1+ cells. It is well-known that different histological patterns are more commonly observed from certain biopsy localizations, e.g., dermatopatic lymphadenopathy is often observed in inguinal and axillary lymph nodes, supposedly due to drainage of large areas of the skin at these localizations [34]. To the best of our knowledge, it has not previously been described that IDO1 is expressed to different degrees depending on the anatomical site of the lymph node biopsy. That IDO1 is differently expressed could maybe be explained by the fact that lymph nodes in different regions are more commonly exposed to certain antigens [1], and thus the immune response, including activity of the kynurenine pathway, is different at different anatomical sites in benign lymphadenopathies. Additionally, in a previous study from our research group of patients with diffuse large B-cell lymphoma (DLBCL), we found that the high expression of MYC and BCL2 by IHC was more often observed in patients with abdominal lymph node involvement, compared to patients without abdominal lymph node involvement [35]. Thus, the localization of the lymph node in lymphomas and benign lymphadenopathies is something that may be taken into account when different IHC markers are investigated in future studies. 

### 4.4. Expression of PD-L2 and PD-L1 

In a comprehensive study on different lymphomas, PD-L1 was more often expressed than PD-L2 in lymphomas, while certain lymphomas showed the expression of PD-L2 but not PD-L1 in some cases of DLBCL and primary mediastinal B cell lymphoma [36]. Interestingly, in our cohort, there were no significant correlations between PD-L2+ cells and IDO1+ or EBV+ cells. The notion that PD-L2 is more restricted to APCs and not induced upon cytokine stimulation to the same degree as PD-L1 in peripheral tissue [14] is somewhat supported by our findings. However, a high amount of follicular PD-L2+ cells was more commonly observed in lymph nodes with follicular hyperplasia. This was supposedly due to expanded GCs, with an abundance of antigen-presenting cells in GCs with follicular hyperplasia, compared to other histologic subtypes [1]. 

### 4.5. PD-L2 and Subsequent Malignancies 

We observed a slightly higher prevalence of cases with high amounts of follicular PD-L2+ cells that subsequently developed malignancies, however not to a statistically significant degree (*p* = 0.09). The association could be due to the immunosuppressive activity of PD-L2+ [37], which could result in an impaired immune response toward pre-malignant or malignant cells in patients with a highly active PD-1 pathway. However, these results should be interpreted with great caution, given the low number of cases with subsequent malignancies (n = 7).

### 4.6. Strengths and Weaknesses 

A major strength with our study is the use of IHC on whole tissue slides instead of tissue microarrays, where there is a substantial risk that we might have missed the infrequent cases with single GCs containing IDO1+ cells. Previous studies used different methods in order to determine the activity of the kynurenine and PD-1 pathway in patients with AD, including measurements of serum [9,15] and lavage from lungs [30]. Our method to stain lymph nodes with IHC for IDO1 and PD-L2 in benign lymphadenopathies has to the best of our knowledge not previously been performed. However, to imunohistochemically stain for IDO1 and PD-L2 in benign lymphadenopathies might not capture the complex inflammatory process in patients with AD. 

## 5. Conclusions

To better understand the immune response in patients with benign lymphadenopathies, we investigated patients with and without AD for IDO1 and PD-L2. We found that: (1) IDO1 was associated with EBV expression; (2) IDO1 was expressed by endothelial cells in a patient with sarcoidosis; (3) IDO1 was more often observed in cervical and axillary lymph nodes; and (4) PD-L2 was more frequently observed in lymph nodes with follicular hyperplasia. Our results further adds important clues to the complex immune response orchestrated by the lymphoid system in benign lymphadenopathies and can be of guidance for future studies on inflammatory and malignant conditions when IDO1 and PD-L2 are studied.

## Figures and Tables

**Figure 1 biomolecules-13-00240-f001:**
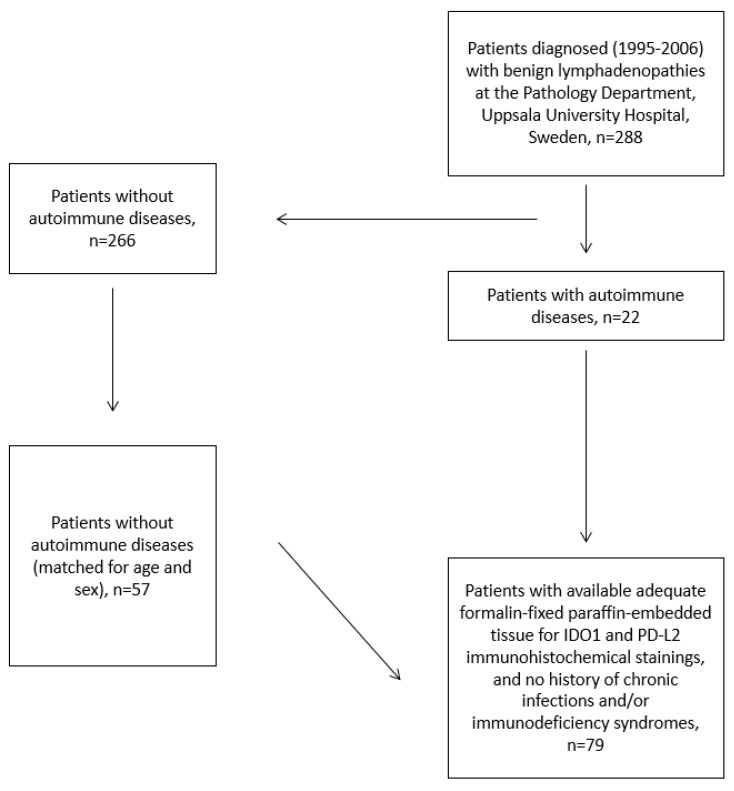
Flowchart of patients included in the study.

**Figure 2 biomolecules-13-00240-f002:**
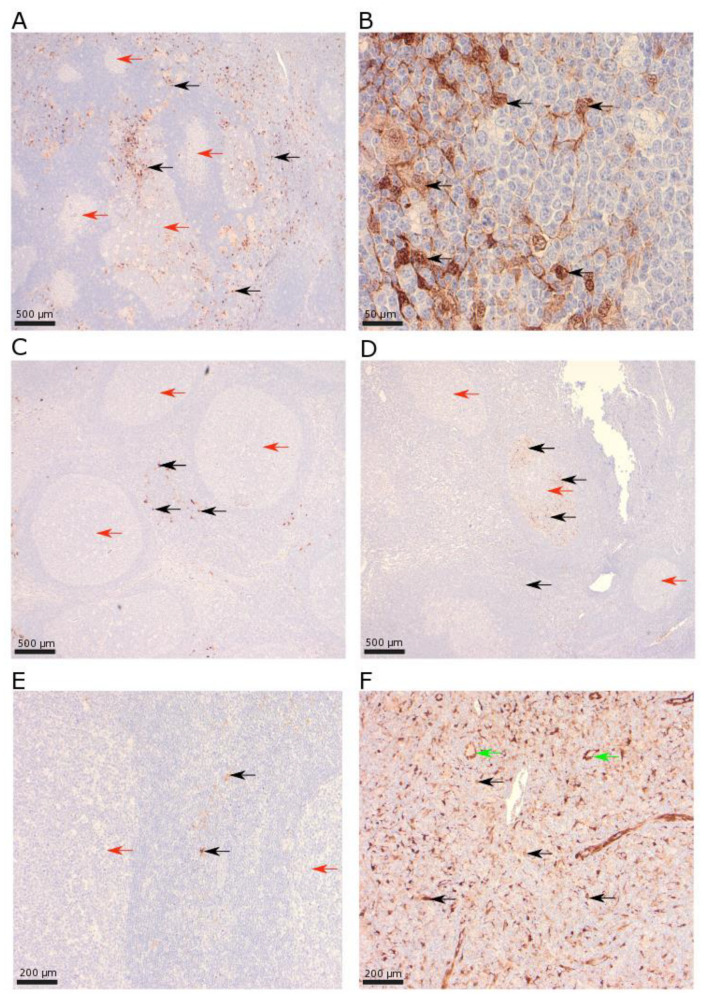
IDO1 in benign lymphadenopathies (rabbit polyclonal antibody HPA023149 (Atlas Antibodies AB, Stockholm, Sweden)) from patients with autoimmune diseases (**C**,**F**) and without autoimmune diseases (**A**,**B**,**D**,**E**). (**A**,**B**) Lymph node with high amounts of IDO1+ cells in both follicular (20%) and interfollicular (20%) areas (magnification 40× and 400×, respectively). (**C**) Low amount of follicular (0%) and intermediate amount of interfollicular (5%) IDO1+ cells (magnification 40×). (**D**) Intermediate amount of interfollicular (5%) and high amount of follicular (10%) IDO1+ cells in one germinal center, with no IDO1+ cells in other germinal centers (magnification 40×). (**E**) Low amounts of IDO1+ cells in both follicular (0%) and interfollicular (2%) areas (magnification 100×). (**F**) High amount of follicular (5%) and a very high amount of interfollicular (75%) IDO1+ cells. Note also endothelial IDO1+ cells (magnification 100×). IDO1+ leukocytes = black arrow, follicular areas = red arrow, IDO1+ endothelial cells = green arrow.

**Figure 3 biomolecules-13-00240-f003:**
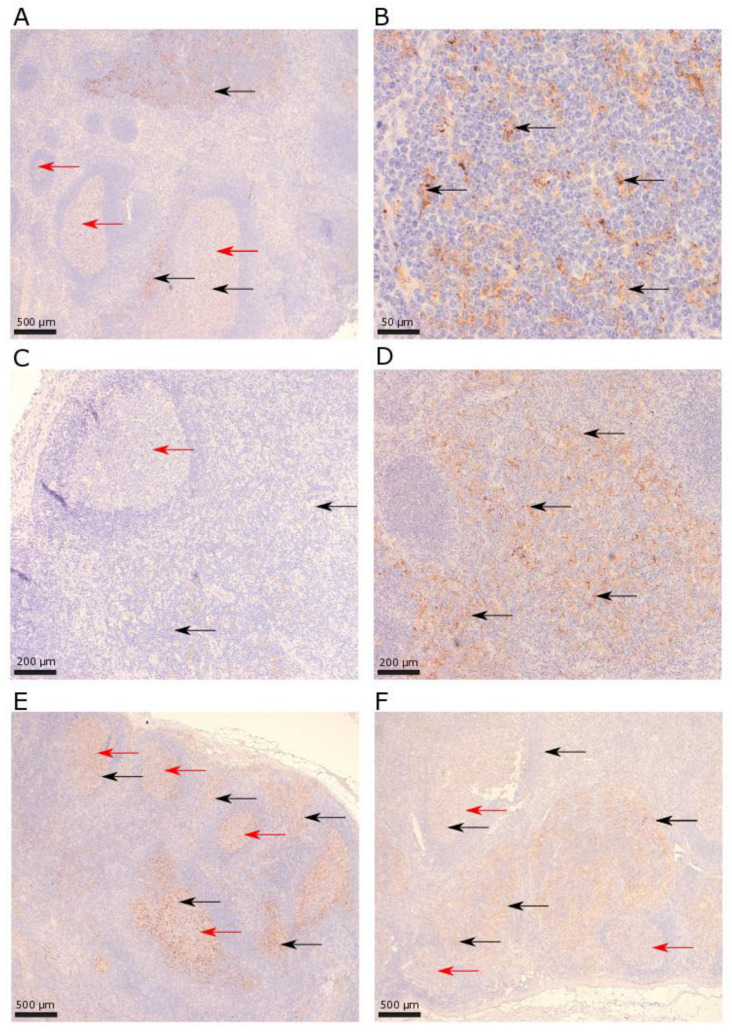
PD-L2 in benign lymphadenopathies (rabbit monoclonal antibody D7U8C/82723 (Cell Signaling Technologies, Danvers, MA, USA)) from patients with autoimmune diseases (**A**,**B**) and without autoimmune diseases (**C**–**F**). (**A**,**B**) Lymph node with high amounts of PD-L2+ cells in both follicular (10%) and interfollicular (14%) areas (magnification 40× and 400×, respectively). (**C**) Low amount of follicular (2%) and interfollicular (2%) PD-L2+ cells (magnification 100×). (**D**) Intermediate amount of follicular (5%) and very high amount of interfollicular (25%) PD-L2+ cells (magnification 100×). (**E**) Very high amount of follicular (20%) and high amount of interfollicular (10%) PD-L2+ cells (magnification 40×). (**F**) Intermediate amount of follicular (5%) and very high amount of interfollicular (30%) PD-L2+ cells (magnification 40×). PD-L2+ leukocytes = black arrow, follicular areas = red arrow.

**Figure 4 biomolecules-13-00240-f004:**
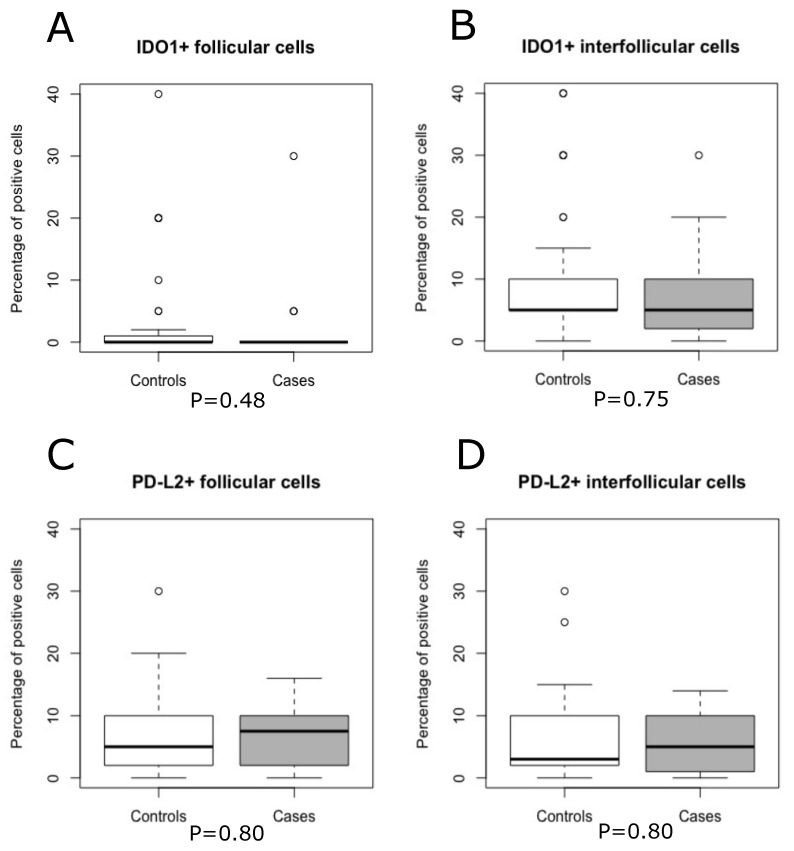
Boxplots of amounts of IDO1+ and PD-L2+ cells in patients with autoimmune diseases (cases) and without autoimmune diseases (controls). Mann–Whitney U-test based *p* values (cases compared with controls).

**Table 1 biomolecules-13-00240-t001:** Clinicopathological characteristics described according to median proportions of IDO1+ and PD-L2+ cells.

Variable	Interfollicular IDO1+ Cells ≥ 5%, n (%) n = 58	Interfollicular IDO1+ Cells < 5%, n (%) n = 20	*p* Value *	Follicular IDO1+ Cells ≥ 1% n = 17	Follicular IDO1+ Cells < 1% n = 54	*p* Value **	Interfollicular PDL2+ Cells ≥ 5%, n (%) n = 40	Interfollicular PDL2+ Cells < 5%, n (%) n = 38	*p* Value ***	Follicular PDL2+ Cells ≥ 5% n = 44	Follicular PDL2+ Cells < 5% n = 26	*p* Value ****
Age			0.93			0.66			0.99			0.76
<60 years	38 (66)	14 (70)		10 (59)	37 (69)		27 (68)	26 (68)		30 (68	16 (62)	
≥60 years	20 (34)	6 (30)		7 (41)	17 (31)		13 (33)	12 (32)		14 (32)	10 (38)	
Sex			0.19			0.36			0.82			0.54
Male	26 (45)	13 (65)		11 (65)	26 (48)		21 (53)	18 (47)		25 (57)	12 (46)	
Female	32 (55)	7 (35)		6 (35)	28 (52)		19 (48)	20 (53)		19 (43)	14 (54)	
Autoimmune disease			0.95			0.53			0.38			0.99
Yes	15 (26)	6 (30)		3 (18)	15 (28)		13 (33)	8 (21)		11 (25)	7 (27)	
No	43 (74)	14 (70)		14 (82)	39 (72)		27 (68)	30 (79)		33 (75)	19 (73)	
Missing	0 (0)	0 (0)		0 (0)	0 (0)		0 (0)	0 (0)		0 (0)	0 (0)	
RA, SLE, and/or Sjögren’s syndrome			0.73			0.10			0.83			0.58
Yes	9 (16)	4 (20)		0 (0)	10 (19)		7 (18)	5 (13)		5 (11)	5 (19)	
No	49 (84)	16 (80)		17 (100)	44 (81)		33 (83)	33 (87)		39 (89)	21 (81)	
Missing	0 (0)	0 (0)		0 (0)	0 (0)		0 (0)	0 (0)		0 (0)	0 (0)	
Steroid and/or methotrexate treatment			0.75			0.43			0.61			0.99
Yes	9 (16)	4 (18)		1 (6)	9 (17)		8 (20)	5 (13)		6 (14)	4 (15)	
No	49 (84)	18 (82)		16 (94)	45 (83)		32 (80)	33 (87)		38 (86)	22 (85)	
Missing	0 (0)	0 (0)		0 (0)	0 (0)		0 (0)	0 (0)		0 (0)	0 (0)	
Histopathological pattern *****			0.36			0.24			0.24			<0.001
Follicular	19 (33)	7 (35)		6 (35)	20 (37)		13 (33)	13 (34)		24 (55)	2 (7)	
Paracortical	15 (26)	4 (20)		3 (18)	16 (30)		10 (25)	10 (26)		8 (18)	11 (42)	
Histiocytosis	9 (16)	3 (15)		3 (18)	9 (17)		5 (13)	7 (18)		4 (9)	7 (27)	
Granulomatous	1 (2)	0 (0)		0 (0)	1 (2)		1 (3)	0 (0)		0 (0)	1 (4)	
Necrosis	1 (2)	0 (0)		0 (0)	0 (0)		0 (0)	1 (3)		0 (0)	0 (0)	
Sclerosis	1 (2)	0 (0)		0 (0)	0 (0)		1 (3)	1 (3)		0 (0)	0 (0)	
Piringer	2 (3)	0 (0)		2 (12)	0 (0)		0 (0)	2 (5)		1 (2)	1 (4)	
Unremarkable	9 (16)	2 (10)		3 (18)	5 (9)		9 (23)	2 (5)		5 (11)	3 (12)	
Unclassifiable	1 (2)	4 (20)		0 (0)	3 (6)		1 (3)	3 (8)		2 (5)	1 (4)	
Lymph node localization			0.03			0.54			0.39			0.43
Cervical	24 (41)	5 (26)		9 (53)	18 (34)		16 (40)	13 (34)		19 (43)	7 (27)	
Axilla	18 (31)	3 (16)		4 (24)	13 (25)		12 (30)	8 (21)		10 (23)	8 (31)	
Inguinal	12 (21)	6 (32)		4 (24)	3 (6)		7 (18)	11 (29)		10 (23)	7 (27)	
Mediastinum	1 (2)	2 (11)		0 (0)	3 (6)		1 (3)	3 (8)		1 (2)	2 (8)	
Abdomen	2 (3)	0 (0)		0 (0)	2 (4)		0 (0)	2 (5)		1 (2)	1 (4)	
Breast	0 (0)	1 (5)		0 (0)	1 (2)		1 (3)	0 (0)		1 (2)	0 (0)	
Face	1 (2)	1 (5)		0 (0)	1 (2)		1 (3)	1 (3)		1 (2)	0 (0)	
Leg	0 (0)	1 (5)		0 (0)	1 (2)		1 (3)	0 (0)		1 (2)	0 (0)	
Unknown	0 (0)	1 (5)		0 (0)	1 (2)		1 (3)	0 (0)		0 (0)	0 (0)	
Subsequent malignancy			0.99			0.99			0.70			0.09
Yes	5 (9)	2 (10)		2 (12)	5 (9)		3 (8)	4 (11)		2 (5)	5 (19)	
No	53 (91)	18 (90)		15 (88)	48 (89)		37 (93)	33 (87)		41 (93)	21 (81)	
Missing	0 (0)	0 (10)		0 (0)	1 (2)		0 (0)	1 (3)		1 (2)	0 (0)	

IDO1 = Indoleamine 2,3 dioxygenase; PD-L2 = Programmed death ligand 2; MTX = Methotrexate; RA = Rheumatoid arthritis; SLE = Systemic lupus erythematosus; EBV = Epstein–Barr Virus, * Comparing interfollicular IDO1+ cells ≥ 5% with interfollicular IDO1+ cells < 5%, ** Comparing follicular IDO1+ cells ≥ 1% with follicular IDO1+ cells < 1%, *** Comparing interfollicular PD-L2+ cells ≥ 5% with interfollicular PD-L2+ cells < 5%, **** Comparing follicular PD-L2+ cells ≥ 5% with follicular PD-L2+ cells < 5%, ***** According to predominant pattern.

**Table 2 biomolecules-13-00240-t002:** Correlation of interfollicular and follicular cells in cases and controls. Spearman rho correlation coefficient and *p* value.

	Follicular IDO1+ Cells	Interfollicular PD-L2+ Cells	Follicular PD-L2+ Cells	Interfollicular PD-1+ Cells	Follicular PD-1+ Cells	Interfollicular PD-L1+ Cells	Follicular PD-L1+ Cells	Interfollicular EBV+ Cells	Follicular EBV+ Cells
Interfollicular IDO1+ cells	0.39 <0.001	−0.01 0.92	0.005 0.97	−0.15 0.19	−0.16 0.18	0.30 0.007	0.16 0.18	0.32 0.004	0.05 0.66
Follicular IDO1+ cells		−0.09 0.48	0.18 0.15	−0.12 0.31	0.09 0.48	0.28 0.02	0.22 0.06	0.34 0.004	0.18 0.12
Interfollicular PD-L2+ cells			0.19 0.12	0.10 0.39	−0.04 0.74	0.10 0.39	−0.06 0.65	−0.10 0.36	0.01 0.95
Follicular PD-L2+ cells				0.20 0.09	0.15 0.21	0.08 0.49	0.37 0.002	0.16 0.19	0.21 0.09

IDO1 = Indoleamine 2,3 dioxygenase; PD-L2 = Programmed death ligand 2; PD-1 = Programmed death receptor 1; EBV = Epstein–Barr virus.

## Data Availability

Data in this study can be provided upon reasonable request.

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
