# Peer review of "Expression of IDO1 and PD-L2 in Patients with Benign Lymphadenopathies and Association with Autoimmune Diseases"

_biomolecules, 2023, doi:10.3390/biom13020240_

Round 1

Reviewer 1 Report

I read with great interest the manuscript by Abdulla et al. The manuscript is well written and important in its field of research. The following observations are, however, made:

·       The Authors should better explain the scoring system adopted to quantify the immunoreactivity of both markers IDO1 and PD-L2. This significantly impacts on the statistical analysis, subjectivity and biases, and scientific value of the entire study.

·       Why the Authors do not apply a more robust computer-aided image analysis system?

·       Minor orthographical and grammatical errors have been found throughout the manuscript.

Author Response

I read with great interest the manuscript by Abdulla et al. The manuscript is well written and important in its field of research. The following observations are, however, made:

We appreciate this comment from the reviewer, that our manuscript is of great interest for the readers of Biomolecules.

·       The Authors should better explain the scoring system adopted to quantify the immunoreactivity of both markers IDO1 and PD-L2. This significantly impacts on the statistical analysis, subjectivity and biases, and scientific value of the entire study.

Answer: We agree with the reviewer that it is of utmost importance to use a method to evaluate immunohistochemical stainings that is standardized and reproducible in order to obtain reliable results. The description of how we evaluated the stainings in the previous version of the manuscript was not correct and it did not fully encompass how elaborate we were during our evaluation of the material, we apologize for this error, which was caused by a slight misunderstanding between the first author who evaluated the material, and the last author who wrote the manuscript. The amount of positive cells in follicular and interfollicular areas was not only semi-quantitatively assessed, the amount of positive cells was calculated in representative areas of the lymph node. For each case and staining, 5 representative high power fields (HPF) for follicular and 5 representative HPF for interfollicular IDO1+ and PD-L2+ cells were chosen and quantified at 400x magnification. The material was evaluated by an experienced hematopathologist, which is highly familiar with evaluating immunohistochemical stainings both in research and in routine clinical settings. In addition, and in line with the comment from the editor, we have also clarified which cells we considered as positive, i.e. cells with at least weak staining intensity and we did not further grade the staining intensity during our evaluation. We have added all this information to the materials and methods section.

·       Why the Authors do not apply a more robust computer-aided image analysis system?

Answer: We have used image analysis in previous studies from our research group, when we evaluated immunohistochemical stainings in different lymphomas on tissue microarrays (TMA). The advantage of using image analysis on TMA is that we can study several cases on a single slide. We used whole slides in the present study, and thus would be required to digitally scan more than 150 slides, which would be laborsome. There are recent studies using image analysis on whole slides, e.g. by the German Hodgkin Study Group (Whole-slide image analysis of the tumor microenvironment identifies low B-cell content as a predictor of adverse outcome in patients with advanced-stage classical Hodgkin lymphoma treated with BEACOPP, Jachimowicz et al, 2021, Haematologica), but in this study the overall cellularity of the tumor engaged lymph node was considered. We believe that it would not be possible with our current image analysis software (Visiomorph) to recognize and calculate separate expression of positive cells in follicular and interfollicular areas of the lymph nodes. Thus, we would be required to manually mark every follicular and interfollicular area of the lymph node for each slide, which would be too time-consuming for this project.

·       Minor orthographical and grammatical errors have been found throughout the manuscript.

Answer: Minor language errors have been corrected in the revised manuscript.

Reviewer 2 Report

Biomolecules-2096431

Expression of IDO1 and PD-L2 in patients with benign lymphadenopathies and association with autoimmune diseases 

The original article “Expression of IDO1 and PD-L2 in patients with benign lymphadenopathies and association with autoimmune diseases" (Biomolecules-2096431) by Abdulla M, et al. demonstrated that expression IDO1 and PD-L2 in lymph nodes in the patients with autoimmune disease (AD). The author previously reported that PD-1 expression was high and PD-L1 and EBV expressions were low benign lymph nodes in the patients with AD. This original article was follow-up data about IHC staining in lymph nodes in the patients with AD. This article is very interesting and contributes to understanding immune environment in benign lymph nodes in the patients with AD. I considered that this article was suitable for acceptance for publication of “biomolecules”, but there were several minor issues for the purpose of improvement for this review article as below.

1.      For the purpose of readers’ better understanding immune microenvironment in lymph nodes among the patients with AD, the author could add more explanation about difference between PD-L1 and PD-L2 in introduction. In addition, it is better to add figure abstract for this article.

2.      Are there difference of expression of IDO1 and PD-L2 among several kinds of AD?

3.      The author demonstrated that the IDO1 expression is high in cervical and axillary lymph nodes compared to another sites of lymph nodes. Is it common that immune system is different among several site of lymph nodes?  

Author Response

Expression of IDO1 and PD-L2 in patients with benign lymphadenopathies and association with autoimmune diseases 

The original article “Expression of IDO1 and PD-L2 in patients with benign lymphadenopathies and association with autoimmune diseases" (Biomolecules-2096431) by Abdulla M, et al. demonstrated that expression IDO1 and PD-L2 in lymph nodes in the patients with autoimmune disease (AD). The author previously reported that PD-1 expression was high and PD-L1 and EBV expressions were low benign lymph nodes in the patients with AD. This original article was follow-up data about IHC staining in lymph nodes in the patients with AD. This article is very interesting and contributes to understanding immune environment in benign lymph nodes in the patients with AD. I considered that this article was suitable for acceptance for publication of “biomolecules”, but there were several minor issues for the purpose of improvement for this review article as below.

 We greatly appreciate this analysis from the reviewer and that our manuscript is suitable for publication in Biomolecules after suggested revisions.

  1. For the purpose of readers’ better understanding immune microenvironment in lymph nodes among the patients with AD, the author could add more explanation about difference between PD-L1 and PD-L2 in introduction. In addition, it is better to add figure abstract for this article.

Answer: We appreciate this suggestion from the reviewer; we have moved part of the discussion on differences between PD-L1 and PD-L2 to the introduction for the purpose of the readers of the manuscript. In addition, we have added a figure abstract.

  1. Are there difference of expression of IDO1 and PD-L2 among several kinds of AD?

Answer: This is an excellent suggestion from the reviewer and we have considered this in supplementary analyses. However, there are few patients with other AD than rheumatoid arthritis (RA), which impairs comparisons between different ADs. We have tried to make a comparison between patients with RA (n=10) and other AD (n=12), but found no differences in expression of IDO1 or PD-L2. We also grouped RA and/or SLE (n=12), RA and/or Sjögren’s syndrome (n=11) and RA, SLE and/or Sjögrens syndrome (n=13) and compared expression of IDO1 and PD-L2 with patients with other AD, but found no statistical significant differences. This has been added as a supplementary table (supplementary Table 1). In addition, previously we did not note in the results that only 3 patients with AD expressed IDO1 in follicular areas, of which 2 patients had sarcoidosis and one patient had autoimmune hepatitis, this is added to the results section. As previously described, the only lymph node in which endothelial cells expressed IDO1 was in a patient with sarcoidosis.

  1. The author demonstrated that the IDO1 expression is high in cervical and axillary lymph nodes compared to another sites of lymph nodes. Is it common that immune system is different among several site of lymph nodes?  

Answer: As stated in the discussion, it is previously described that dermatopatic lymphadenopathy more commonly are observed in inguinal and axillary lymph nodes. To our knowledge, it is not previously described that IDO1 (or PD-L2) is expressed to different degrees at different anatomical sites in benign lymphadenopathies, which is a novel findings with our study, this has been clarified in the discussion.